# Heat Shock Protein DnaJ in *Pseudomonas aeruginosa* Affects Biofilm Formation via Pyocyanin Production

**DOI:** 10.3390/microorganisms8030395

**Published:** 2020-03-12

**Authors:** Bo Zeng, Chong Wang, Pansong Zhang, Zisheng Guo, Lin Chen, Kangmin Duan

**Affiliations:** 1Key Laboratory of Resource Biology and Biotechnology in Western China, Ministry of Education, Northwest University, Xi‘an, Shaanxi 710069, China; zengbo6201@gmail.com (B.Z.); wangchong0910@stumail.nwu.edu.cn (C.W.); zps102065106@stumail.nwu.edu.cn (P.Z.); zishengguo@nwu.edu.cn (Z.G.); 2Department of Oral Biology & Medical Microbiology & Infectious Diseases, Rady Faculty of Health Sciences, University of Manitoba, 780 Bannatyne Ave., Winnipeg, MB R3E 0W2, Canada

**Keywords:** heat shock protein (HSP), pyocyanin (PYO), extracellular DNA (eDNA), virulence, *Pseudomonas aeruginosa*

## Abstract

Heat shock proteins (HSPs) play important biological roles, and they are implicated in bacterial response to environmental stresses and in pathogenesis of infection. The role of HSPs in *P. aeruginosa,* however, remains to be fully elucidated. Here, we report the unique role of HSP DnaJ in biofilm formation and pathogenicity in *P. aeruginosa*. A *dnaJ* mutant produced hardly any pyocyanin and formed significantly less biofilms, which contributed to decreased pathogenicity as demonstrated by reduced mortality rate in a *Drosophila melanogaster* infection model. The reduced pyocyanin production in the *dnaJ* mutant was a result of the decreased transcription of phenazine synthesis operons including *phzA1*, *phzA2*, *phzS*, and *phzM*. The reduction of biofilm formation and initial adhesion in the *dnaJ* mutant could be reversed by exogenously added pyocyanin or extracellular DNA (eDNA). Consistent with such observations, absence of *dnaJ* significantly reduced the release of eDNA in *P. aeruginosa* and addition of exogenous pyocyanin could restore eDNA release. These results indicate *dnaJ* mutation caused reduced pyocyanin production, which in turn caused the decreased eDNA, resulting in decreased biofilm formation. DnaJ is required for pyocyanin production and full virulence in *P. aeruginosa*; it affects biofilm formation and initial adhesion via pyocyanin, inducing eDNA release.

## 1. Introduction

Heat shock proteins (HSPs) were originally reported to recognize consensus DNA binding sites and activate genes encoding protein chaperones in response to elevated temperatures [1]. However, it has become apparent that HSPs participate in gene regulation not only by protecting cellular proteins from environmental stress but also by maintaining a cellular protein homeostasis [2]. HSPs can be divided into six main families according to their molecular weight: HSP100, HSP90, HSP70, HSP60, HSP40 and small heat shock proteins [3]. DnaJ is a typical member of the HSP40 family, and acts as a co-chaperone for HSP70 proteins, which are referred to as master players in protein homeostasis [4,5]. DnaK has been to be a DnaJ interaction partner in many bacteria. DnaJ promotes the release of substrates by promoting the transfer of substrates and stimulating the ATPase activity of its co-chaperone DnaK [6]. DnaJ, together with DnaK and DnaE, can act on the entire *Escherichia coli* proteome because most proteins contain multiple DnaK and DnaJ binding sites [7]. There is also increasing evidence that DnaJ can independently function with other proteins in *Streptococcus pneumoniae* [8]. While HSPs help maintain cellular protein homeostasis, and hence affect physiological activities in general, there is increasing evidence that HSPs play an important role in bacterial pathogenicity [2]. Some heat-shock proteins have been identified as important virulence factors in bacterial pathogens, while others appear to affect pathogenesis indirectly [9].

*Pseudomonas aeruginosa*, a Gram-negative bacterium, is a clinically important opportunistic pathogen that can cause serious infections in humans [10]. *P. aeruginosa* is able to produce a variety of virulence factors that enable it to invade and damage the host, or to evade the host immune system. *P. aeruginosa* secretes many pigmented phenazine compounds that are involved in host immune response evasion [11]. The most well-studied pigmented phenazine compound is pyocyanin (PYO), which is an electrochemically active metabolite and a signal involved in gene regulation and maintaining fitness of bacterial cells [12]. Biofilm formation is one of the most important virulence factors in *P. aeruginosa.* Biofilms protect the bacterium from antimicrobial agents and the host immune system [13]. Both PYO production and biofilm formation in *P. aeruginosa* are controlled by quorum sensing (QS) systems. *P. aeruginosa* possesses several major QS systems, such as the *las*, *rhl*, and *pqs* systems [14], which use N-(3-oxododecanoyl)-homoserine lactone (OdDHL) [15], N-butanoyl-L-homoserine lactone (BHL) [16], and 2-heptyl-3-hydroxy-4(1H) -quinolone (PQS) [17] as their main autoinducers respectively. The hierarchic, acyl-homoserine lactone (AHL) based *las* and *rhl* systems control about 6% of the genes in the genome [18], and, together with the *pqs* system, regulate both PYO production and biofilm formation [19]. Extracellular DNA (eDNA) is one of the most important components in biofilms, and serves as an adhesive, promoting biofilm formation [20]. eDNA release in *P. aeruginosa* is regulated by both QS-dependent and QS-independent mechanisms [21].

Despite the important roles of HSPs in bacterial physiology and their potentially significant roles in pathogenicity, very limited information is available regarding the function of DnaJ in *P. aeruginosa* or its relationship with pathogenicity. In this report, we describe the dramatically reduced PYO production and biofilm formation in a *dnaJ* mutant of *P. aeruginosa* and the potential underlying mechanisms. We present results showing that the reduced PYO production caused by *dnaJ* mutation decreases eDNA release, resulting in decreased biofilm formation. Genetic and biochemical evidence indicating that DnaJ contributes to the virulence of *P. aeruginosa* is discussed.

## 2. Results

### 2.1. Disruption of dnaJ Does Not Affect P. aeruginosa Survival under Different Stress Conditions

A transposon mutant previously obtained in our transposon mutant library showed an absence of pigment in its colony. Analysis of the DNA sequence flanking the transposon insertion site indicates that the PA4760 gene was disrupted (Appendix A) [22]. This encodes DnaJ that belongs to the HSP superfamily. To examine whether the disruption of *dnaJ* (named *dnaJ-M*) could affect the growth of *P. aeruginosa* in response to different stress stimuli, we tested the survival of *P. aeruginosa* and the *dnaJ* mutant after treatment in different conditions, such as heat shock at 50 °C or 60 °C for 1 min, and treatment with 0.1% H_2_O_2_. The colony-forming units (CFU) were counted. Although it has been reported that the mutation of *dnaJ* affects the survival of other bacteria [23], our results showed that the growth and survival of wild-type *P. aeruginosa* PAO1 and a *dnaJ-M* did not differ (Figure 1). The CFU counts after treatment under different stress conditions showed the same result. Therefore, the absence of *dnaJ* in *P. aeruginosa* uniquely had no effect on bacterial growth or survival under the stress conditions tested.

### 2.2. Production of Pyocyanin Was Drastically Reduced in the Absence of dnaJ

PYO is a critical virulence factor secreted by *P. aeruginosa*. It was noticed that the *dnaJ-M* culture had significantly decreased pigment; therefore, the production of pyocyanin in *P. aeruginosa* in the absence of *dnaJ* was measured. The results showed that the mutant produced hardly any PYO, with the level only slightly above the control. Numerically, the wild type produced eight times more PYO than the *dnaJ* mutant (Figure 2a).

In *P. aeruginosa,* phenazine compounds are synthesized by enzymes encoded in several operons. To address the underlying mechanism of the PYO reduction in the *dnaJ* mutant, we first examined the transcriptional levels of the genes associated with phenazine synthesis. The results showed that the expression of *phzM* and *phzS*, which encode putative phenazine-specific methyltransferase and flavin-containing monooxygenase, respectively (enzymes which control the synthesis of PYO from phenazine-1-carboxylic acid (PCA)), was reduced fivefold and twofold respectively (Figure 2b). In addition, the expression of the *phzA1* and *phzA2* operons was also decreased 1.7-fold and fivefold in the *dnaJ* mutant compared with the wild type (Figure 2b). This result indicates that all the operons associated with phenazine synthesis tested showed significantly decreased expression in the mutant type when compared with the wild type, which demonst
rates that DnaJ affected PYO production at the transcriptional level.

### 2.3. The dnaJ Mutation Impaired Biofilm Formation

As PYO is involved in biofilm formation [24], we tested the biofilm formation of the *dnaJ* mutant. Both the initial stage of adhesion during biofilm formation and the final amount of biofilm formed were compared between the wild type and the *dnaJ* mutant. As shown in Figure 3a,b, both adhesion and final biofilm formation were significantly decreased in the mutant, indicating an important role for DnaJ in biofilm formation in *P. aeruginosa.*

Since motilities play a role in biofilm formation, we further examined the effect of *dnaJ* disruption on *P. aeruginosa* motilities. While the twitching motility was not affected (data not shown), both swarming and swimming motilities were decreased in the *dnaJ* mutant (Figure 3c). The decreased swarming motility was somewhat unexpected, considering the decreased biofilm formation in the mutant. It is known that swarming motility is inversely correlated with biofilm formation in *P. aeruginosa* [25]. Since biofilm formation is modulated by multiple factors, the effect of motility was likely veiled by other effects of DnaJ.

### 2.4. The Full Pathogenicity of P. aeruginosa Requires Functional dnaJ

The finding that DnaJ affected PYO production and biofilm formation prompted us to investigate the pathogenicity of the *dnaJ* mutant. The in vivo pathogenicity of the *dnaJ* mutant was compared with that of the wild type using a *Drosophila melanogaster* feeding infection model. As shown in Figure 4, the survival rate of the fruit flies fed with the *dnaJ* mutant was significantly higher compared with those infected with the wild type. Apparently, the full pathogenicity of *P. aeruginosa* requires functional *dnaJ*.

### 2.5. The Effect of dnaJ on Biofilm Formation Was through PYO and eDNA

Previous studies have shown that PYO can promote eDNA release in *P. aeruginosa* and that eDNA is a major component of biofilms [26]. Because the production of PYO was significantly decreased in the *dnaJ* mutant, we suspected that the changed biofilm production and adhesion in the *dnaJ* mutant could be due to decreased production of eDNA. The eDNA was therefore isolated directly from the cultures of the wild type and the *dnaJ* mutant, and the eDNA amounts were measured and compared. The results show that the *dnaJ* mutant produced much less eDNA compared with the wild type and the complementary strain (Figure 5a).

To confirm that the lessened eDNA in the *dnaJ* mutant was due to the reduced PYO observed in our experiments, we measured the amount of eDNA in supernatants of the *dnaJ* mutant grown with exogenously added PYO. The result showed that eDNA release was restored to the wild type level by the exogenous PYO (Figure 5a). Consistently, biofilm formation and adhesion in the *dnaJ* mutant were also restored by addition of PYO (Figure 5b,c). Similarly, addition of eDNA isolated from the cultures was also able to restore biofilm formation and adhesion. These results confirm that the reduced biofilm formation and initial adhesion by the *dnaJ* mutant was due to the decreased PYO and subsequently reduced eDNA.

### 2.6. DnaJ Affected the Production of PYO through QS Pathways

To address the underlying mechanism of PYO reduction in the *dnaJ* mutant, we examined the QS regulatory systems that play a crucial role in the production of PYO [27]. The expression of the QS genes was measured using the *lux*-based reporter system. The results obtained indicate that, except *lasR*, all the QS-associated genes tested, including *lasI rhlR rhlI* and *pqsA*, were downregulated in the *dnaJ* mutant (Figure 6a). Consistent with this observation, significantly fewer PQS signals were detected in the *dnaJ* mutant (Figure 6b). These results suggest that DnaJ affected the production of PYO through both the AHL- and PQS-based QS pathways.

## 3. Discussion

HSP genes are highly conserved in all eukaryotes and prokaryotes, and they participate in a number of fundamental cellular processes such as controlling protein folding and gene expression. Interestingly, some HSPs have been identified as virulence factors, while others are involved in pathogenesis indirectly [9]. *P. aeruginosa* is an important human pathogen that causes serious infections, especially in patients with underlying medical conditions or immunocompromised patients. The data obtained in this study indicate that the inactivation of *dnaJ* in *P. aeruginosa* caused a dramatic reduction in PYO production, decreased adherence and biofilm formation. HSP DnaJ in *P. aeruginosa* seems to have a significant effect on the pathogenicity of *P. aeruginosa*.

HSPs are important for bacterial survival and adaptation to adverse environmental conditions. Previous studies have shown that the absence of *djlA,* a member of the same DnaJ family as *Legionella dumoffii* [28] and *Streptococcus suis* type 2, can affect bacterial growth [29], but the mutation of *dnaJ* of *P. aeruginosa* has no effect on bacterial growth (Figure 1a). The cells of the *dnaJ* mutant treated with heat shock and H_2_O_2_ had a similar survival rate to the wild type cells. It is possible that *P. aeruginosa* has other HSPs that could compensate the loss of *dnaJ*. For instance, *P. aeruginosa* has *cbpA* on its chromosome; this is a *dnaJ* homolog that has a similar function to *dnaJ* in cell growth and division in *E. coli* [6].

The most obvious change in the *P. aeruginosa dnaJ* mutant was its diminished PYO production. *P. aeruginosa* has the ability to produce pigmented secondary metabolites including PYO and several other phenazine compounds. PYO is a critical virulence factor in *P. aeruginosa* during chronic infection [30]. It appears that the reduction of PYO production was mediated by the downregulation of *phzM*, *phzS*, and the *phzA1* and *phzA2* operons. Therefore, DnaJ may affect PYO production by directly regulating the *phz* genes. However, it is well known that QS systems control the production of PYO in *P. aeruginosa;* this points to an indirect route where DnaJ could assert its effect on PYO production. The reduced expression of QS-related genes, such as *lasI*, *rhlR*, *rhlI*, and *pqsA*, in the *dnaJ* mutant seems to indicate that the DnaJ indirectly affects PYO production through the QS systems. The comparison of PQS in the *P. aeruginosa* wild type and *dnaJ* mutant cultures confirms the effect of DnaJ on the QS systems.

Biofilm formation is an important characteristic during *P. aeruginosa* chronic infection. Bacterial cells in biofilms are protected from antimicrobial drugs and the host immune response. Both initial adhesion and final biofilm formation in the *dnaJ* mutant were significantly decreased. Multiple factors could have contributed to such a reduction in biofilm formation. PYO can promote extracellular DNA (eDNA) release in *P. aeruginosa* and eDNA is essential for biofilm formation [26]. We detected the production of eDNA in the LB and Pseudomonas broth (PB) media with or without exogenous PYO. The results confirmed that the reduced production of eDNA was, at least partially, due to the loss of PYO production. Previous literature has also shown that a *pqsA* mutant, deficient in the production of PQS, released low amounts of extracellular DNA, and a *pqsL* mutant, which overproduces PQS, released large amounts of extracellular DNA [31]. Therefore, the decreased eDNA in the *dnaJ* mutant could also be due to the decreased PQS caused by the absence of *dnaJ*.

Both PYO and biofilms play an important role in the pathogenicity of *P. aeruginosa*. The effect of *dnaJ* on the in vivo pathogenicity of *P. aeruginosa* was confirmed by the fruit fly infection assay. The mutant-strain-fed flies had better survival rates than the wild-type-fed flies. However, it is noted that the effect of *dnaJ* could be through virulence-related pathways other than PYO and biofilm formation. It is possible that other virulence factors not tested in our study could also been affected by the disruption of *dnaJ*. After all, DnaJ as a chaperon protein could interact with many virulence-associated or non-virulence-associated proteins. It has been reported that nearly all the proteins in the *E. coli* proteome are predicted to contain multiple DnaK- and DnaJ-binding sites [7]. Therefore, the effect of DnaJ on pathogenicity in *P. aeruginosa* may be multifactorial. Nevertheless, the requirement of DnaJ for the full pathogenicity of *P. aeruginosa* appears quite clear.

The flagella-propelled swimming and swarming motilities of *P. aeruginosa* are also a virulence factor in host invasion [32]. In agreement with previous studies, which show that *dnaJ* mutation in *E. coli* causes decreased flagellin synthesis [33], the defective swimming and swarming in the *dnaJ* mutant of *P. aeruginosa* might also be caused by defected flagellar biosynthesis. Since swarming motility is known to be related to biofilm formation, the decreased swarming motility in the *dnaJ* mutant seems to contradict the reduced biofilm formation and reduced initial adhesion observed. Decreased motility would promote biofilm formation, especially at the initial stages [34]. However, the reduced biofilm formation is probably a combined result of different changes in the *dnaJ* mutant. The effect of the decreased motility was probably concealed by other changes such decreased eDNA release, that would instead result in decreased biofilm formation.

Taken together, we reported that DnaJ, an HSP40 family protein in *P. aeruginosa*, contributes to the bacterium’s pathogenicity. It significantly affects PYO production and biofilm formation. Further study of DnaJ and its regulation pathway should contribute to a better understanding of the pathogenesis of *P. aeruginosa* infection and to the exploration of new therapeutic targets.

## 4. Materials and Methods

### 4.1. Bacterial Strains, Culture and Media

The plasmids and strains used in this study have are listed in Appendix A and the primers used in this study are listed in Appendix A. For the culture media, Luriae Bertani (LB) medium and Pseudomonas broth (PB) (2% Peptone, 0.14% MgCl_2_, 1% K_2_SO_4_), which is a medium to maximize pyocyanin production in liquid culture, were used. All strains were grown aerobically at 37 °C. The antibiotics used in this study were as follows. For *Escherichia coli*, Gentamicin (Gm, 15 µg/mL), Kanamycin (Kn, 50 µg/mL) and ampicillin (Amp, 100 µg/mL) were used. For *P. aeruginosa*, carbenicillin (Cb, 500 µg/mL) and trimethoprim (Tmp, 300 µg/mL) were used. All antibiotics were purchased from MP Biomedicals, LLC (Solon, OH, USA) and other chemicals were purchased from the Tianjin Kemiou Chemical Reagent Co., Ltd. (Tianjin, China).

### 4.2. Bacterial Survival Test by Growth Assay

The growth of each strain was measured using the microplate reader and colony-forming units (CFU). The cells were cultured overnight at 37 °C and 200 rpm after inoculation in fresh LB broth. The bacterial solution was adjusted from OD_600_ to 0.01, and then the growth was measured in 96-well plates using the microplate reader according to a 5% inoculation. In addition, the strains cultivated in 5 mL LB overnight were adjusted to OD_600_ = 0.5, and a serious of dilutions from 10^−1^ to 10^−7^ were made. Each dilution gradient was placed 5 μL on an LB plate and then incubated for 16 h. The experiment was repeated at least three times.

### 4.3. Motility Assays

The motility assay was carried out as described previously with minor modifications [35]. The swarming motility medium comprised 0.8% nutrient broth, 0.5% glucose and 0.5% agar. The swimming medium comprised 1% tryptone, 0.5% sodium chloride and 0.3% agar. Plates were dried at room temperature overnight before being used. The 2 μL bacterial solution was spotted onto the swimming and swarming plates. After the solution was completely inhaled into the medium, they were carefully removed to a 37 °C incubator for 24 h. Photographs were taken with the LAS-3000 imaging system (Fuji Corp). The experiment was repeated at least three times.

### 4.4. Measurement of PYO Production

Pyocyanin was extracted from culture supernatants and measured using previously reported methods [36]. Briefly, the cells were cultured for 18 h in PB medium at 37 °C and 3 mL chloroform was added to 5 mL culture supernatant. After extraction, the chloroform layer was transferred to a fresh tube and mixed with 1 mL 0.2 M HCl. After centrifugation, the top layer (0.2 M HCl) was removed and its A520 was measured. The amount of pyocyanin, in μg/mL, was calculated using the following formula: A520/A600 × 17.072 = μg of pyocyanin per mL.

### 4.5. Monitoring Gene Expression by Lux-Based Reporters

The reporter strains (named pMS402) carry a promoterless *luxCDABE* reporter gene cluster (p-*lux* reporter system). The promoter region of the target genes was amplified by PCR, and the PCR products were digested by BamHI-XhoI. The enzyme-digested products were cloned into pMS402. The expression of these genes was monitored using light production Luminescent *lux* reporter strains in 96-well microtiter plates as reported previously with minus modification [37]. Briefly, strains containing different reporters were cultivated overnight in LB broth supplemented with Tmp (300 μg/mL). Overnight cultures of the reporter strains were diluted to an optical density at 600 nm (OD_600_) of 0.2 and cultivated for an additional 2 h before use as inoculants. Aliquots of a fresh culture (5 μL) were inoculated into parallel wells on a 96-well black plate with a transparent bottom, which contained 95 μL of medium with Tmp (300 μg/mL). To prevent evaporation during the assay, 70 μL of filter-sterilized mineral oil was added. Both luminescence and bacterial growth (OD_600_) were measured every 30 min for 24 h in a BioTek Synergy 2 plate reader, and data were collected using Gen5 software. The level of light production, measured in counts per second (cps), was proportional to the level of gene expression. The light production values were then normalized to the level of bacterial growth. The level of gene expression is presented as the number of relative expressions, calculated as cps/OD_600_ by use of the normalized cps values.

### 4.6. Quantification of Initial Adhesion and Biofilm Formation

The initial adhesion and biofilm production were measured in a static system, as previously described [38], with minor modifications. Quantification of the initial adhesion was carried out in 15 mL borosilicate tubes. Briefly, cells from overnight cultures were inoculated at 1:100 dilutions into PB media and statically grown at 37 °C for 8 h. For the biofilm production, strains were grown at 37 °C for at least 24 h. After removing the bacterial solution, the test tube was stained by 1% crystal violet for 20 min, and 1mL 95% ethanol was used to dissolve the crystal violet. The solution was measured at 600 nm. The experiment was repeated at least three times.

### 4.7. In Vivo Pathogenicity Assay Using Drosophila Melanogaster Feeding Infection Model

The fly-feeding assay was applied as previously reported [39]. For infection, 1.0 mL of the culture, which was adjusted to OD_600_ = 2.0 using the media the strain was grown in, was collected by centrifugation. The supernatant was then removed and the resulting pellet was resuspended in 100 mL of 5% sucrose. Then, 100 μL 0.1 M sucrose solution and the same amount of bacterial solution were added to filter paper, which was placed on the surface of 2 mL of solidified 0.1 M sucrose agar in a 20 mL glass tube with silica gel stopper. Male flies (3–5 days old) were starved for 3 h before 10 flies were added to each tube. A cold shock method was used for anesthetizing flies throughout the sorting and transferring process. Infection tubes were stored at 26 °C in a humidity-controlled environment. The number of live flies at the start of the experiment was documented, and live flies were counted at 24 h intervals for 6 days. Survival percent was determined for groups of 60 drosophila flies (with 6 batches of 10 flies per vial) at 26 °C.

### 4.8. eDNA Assay

The cells were cultured in 5 mL PB for overnight at 37 °C 200 rpm. The cells from overnight cultures were separated by centrifugation at 6000× *g* for 10 min at 4 °C. After the supernatant was filtered, the cell-free supernatant (3 mL) was used for DNA recovery following the instructions of the DNA gel extraction kit, which are as follows. Briefly, add 1 volumes of buffer to 1 volume of supernatant. To bind DNA, apply the sample to the column and then centrifuge for 1 min. For sample volumes > 800 μL, simply load the remainder and spin again. To wash, add 0.75 mL of washing buffer into the column and centrifuge for 1 min (repeat once). Discard the flow-through and centrifuge the column for an additional 1 min at 7900× *g*. To elute DNA, add 50 μL of buffer EB (10 mM Tris-Cl, pH 8.5) to the center of the membrane, and then centrifuge the column for 1 min. The purified eDNA (15 μL) mixed with the loading buffer is then to be analyzed on a gel. After completing the process, the gel was visualized using the Syngene GeneGenius Bio Imaging System (Synoptics Ltd, Cambridge, UK). The experiment was repeated at least three times.

### 4.9. PQS Extraction and Quantification

Each strain was grown in PB broth for 24 h, and then the cells of supernatant were removed by centrifugation at 20,000× *g* for 5 min, and the supernatant was filtered through a polytetrafluoroethylene membrane (pore size, 0.22 μm). Aliquots of culture supernatants were subjected to two extractions by addition of one volume of acidified ethyl acetate (0.01% acetic acid). The organic phase was transferred to a fresh tube and dried to completion. The solutes were re-suspended in methanol for TLC analysis. The sample was spotted onto a TLC plate which had been previously soaked for 30 min in 5% KH_2_PO_4_ and activated at 100 °C for 1 h. Extracts (PQS) were separated using a dichloromethane: methanol (95:5) system until the solvent front reached the top of the plate. The plate was visualized using a UV and photographed. Standard QS signals were used as a positive control. The experiment was repeated at least three times.

### 4.10. Statistical Analysis

All statistical analyses were performed using GraphPad Prism version 5 (GraphPad Software, La Jolla, CA, USA). The two-tailed unpaired *t*-test was used to analyze the data. * *p* < 0.05; ** *p* < 0.01; *** *p* < 0.001. Values are expressed as mean *±* standard errors of the means.

## Figures and Tables

**Figure 1 microorganisms-08-00395-f001:**
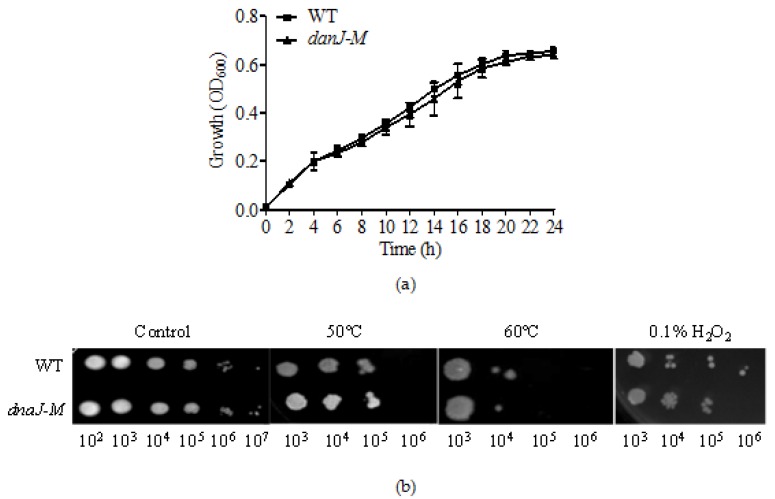
The growth profile of strains in Luria-Bertani (LB) broth and after different treatments. (**a**) Growth comparison in LB broth; (**b**) Survival after heat shock and H_2_O_2_ treatment. Heat shock was carried out at 50 °C and 60 °C for 1 min, and H_2_O_2_ treatment with the final concentration at 0.1% for 1 min. Serial dilutions are shown underneath the images. Untreated was used as a control. The results shown are representative of three independent experiments with similar results.

**Figure 2 microorganisms-08-00395-f002:**
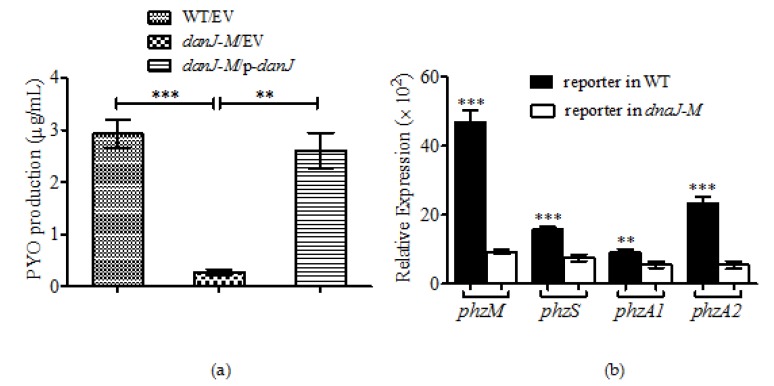
Comparison of pyocyanin production and expression of related genes in the wild-type *dnaJ* mutant and the complementation strains. (**a**) Reduced production of pyocyanin in *dnaJ-M*. The means of three independent experiments are shown. (**b**) The expression of *phzM*, *phzS*, *phzA1* and *phzA2* using p-*lux* promoter-reporter systems in the wild-type (black) and the *dnaJ-M* (white) strains. The values are presented as cps normalized to OD_600_. Data shown are averages from triplicate experiments ± standard errors. A two-tailed unpaired *t*-test was performed using GraphPad software version 5.0 (** *p* < 0.01; *** *p* < 0.001). The results shown are representative of three independent experiments with similar results.

**Figure 3 microorganisms-08-00395-f003:**
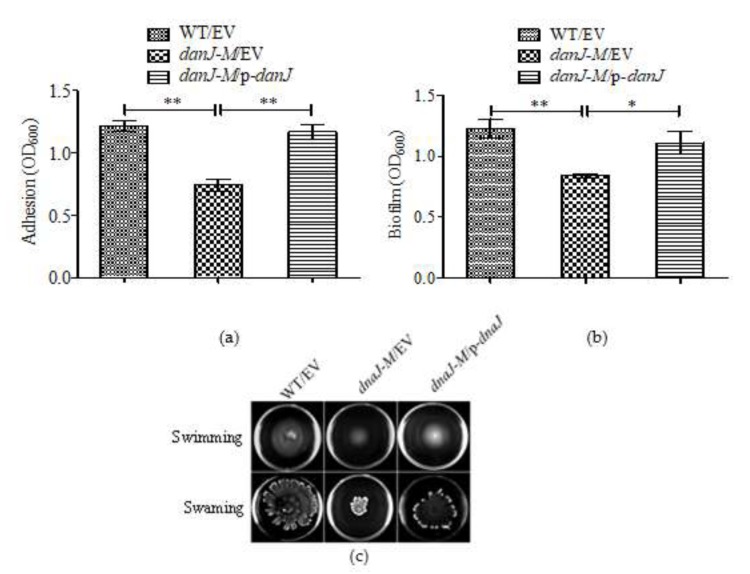
Reduced initial adhesion, final biofilm formation, and motility in *dnaJ-M*. (**a**) The ability of adhesion of wild type PAO1, *dnaJ-M*, and the complementation strains; (**b**) the production of biofilms in these strains; (**c**) the changed swarming and swimming motilities in *dnaJ-M*. The images were captured using a LAS-3000 imager system. EV: Empty vector control. p-*dnaJ*: *dnaJ* complementation construct. A two-tailed unpaired *t*-test was performed using GraphPad software version 5.0 (* *p* < 0.05; ** *p* < 0.01). NS: Not Significant.

**Figure 4 microorganisms-08-00395-f004:**
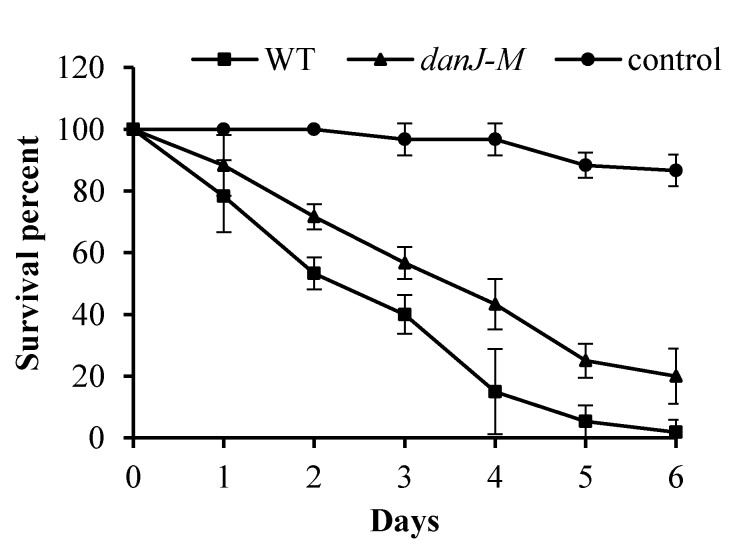
Analysis of *P. aeruginosa* pathogenicity using the *Drosophila melanogaster* infection model. Survival curves of the fruit flies chronically infected with different strains are shown. Square: flies infected with the wild type; Triangles: flies infected with the *dnaJ-M*; Circles: uninfected controls.

**Figure 5 microorganisms-08-00395-f005:**
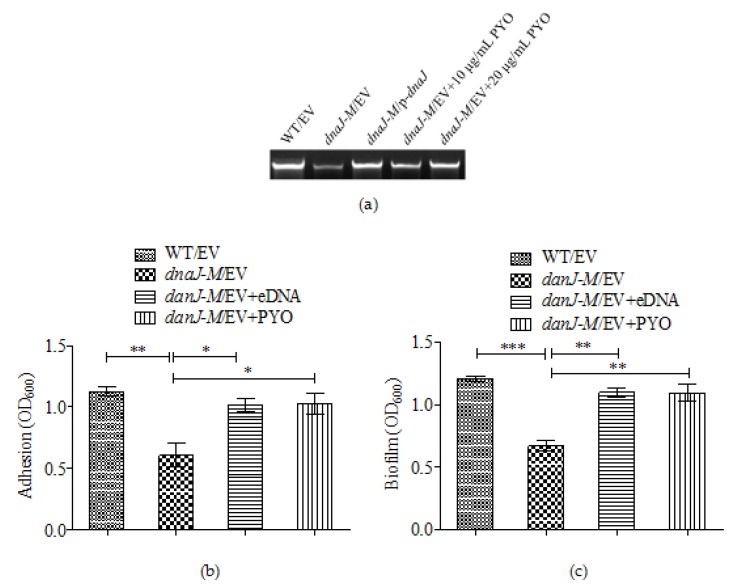
The contribution of eDNA and PYO to adhesion and biofilm production. (**a**) eDNA production in WT, *dnaJ-M*, and complementation strains. eDNA productions of *dnaJ-M* with exogenous pyocyanin at 10 µg/mL and 20 µg/mL (final concentration) are also shown. (**b**) The initial adhesion and (**c**) biofilm formation in WT, *dnaJ-M* and *dnaJ-M* with exogenous eDNA at 2 µg/mL (final concentration) or pyocyanin at 10 µg/mL Averages of triplicate experiments ± standard errors of the means are shown. A two-tailed unpaired *t*-test was performed using GraphPad software version 5.0 (* *p* < 0.05; ** *p* < 0.01; *** *p* < 0.001). EV: empty vector control. p-*dnaJ*: *dnaJ* complementation construct.

**Figure 6 microorganisms-08-00395-f006:**
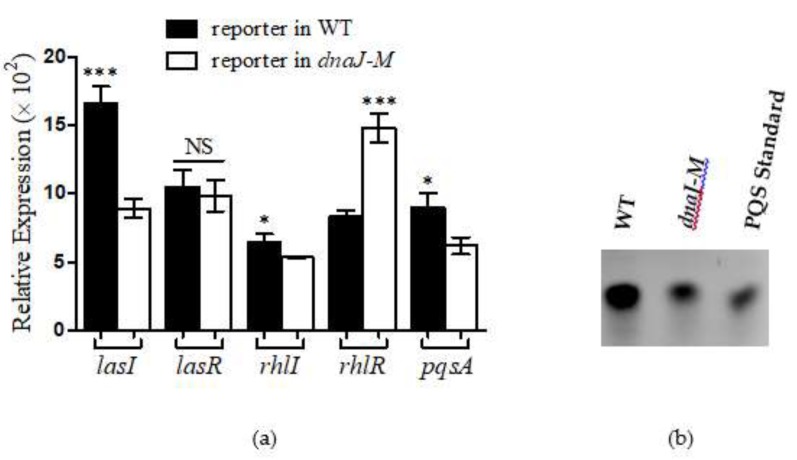
The effect of *dnaJ* on quorum sensing (QS) genes and 2-heptyl-3-hydroxy-4(1H) -quinolone (PQS) signal production. (**a**) The expression of *lasI*, *lasR*, *rhlI*, *rhlR* and *pqsA* was compared between the wild-type, PAO1 (black), and the mutant strains, *dnaJ-M* (white). The expression levels presented are cps-normalized to OD_600_. Averages of triplicate experiments ± standard errors of the means are shown. Two-tailed unpaired *t*-test was performed using GraphPad software version 5.0 (* *p* < 0.05; ** *p* < 0.01; *** *p* < 0.001). NS: Not Significant. (**b**) Comparison of PQS production between the WT and *dnaJ-M*. PQS in the cultures was isolated and analyzed on high performance thin layer chromatography (TLC). Extract from 500 µL of culture equivalent was loaded onto the TLC plate.

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
