# Peer review of "Heat Shock Protein DnaJ in Pseudomonas aeruginosa Affects Biofilm Formation via Pyocyanin Production"

_microorganisms, 2020, doi:10.3390/microorganisms8030395_

Round 1
Reviewer 1 Report
The authors proved that the HSP dnaJ has a role in the biofilm formation and the pathogenicity of P aeruginosa. They proved that a dnaJ mutation will not affect the survival, but will decrease the production of pyocyanine (through QS pathways), the biofilm formation (through PYO and eDNA) and the full pathogenicity of P. aeruginosa require a functional dnaJ.
The paper is well written following all steps needed to prove the hypothesis.
The "Introduction" presents the known roles of HSP, with special attention to dnaJ and P.aeruginosa. here -just a spelling error in line 2: P.aeruginosa is ability... is able
The results are presented in a clear order and a way that permits the reader to follow and to understand well the idea of this research. All the needed steps of the research are also proved by the figures presented and explained. The aim of the research was followed and proved clearly and the conclusions of the study are fully supported by the facts.
In the "Discussion" chapter, the authors explained clearly the results in relation to other data published in this field.
The "Material and methods" chapter presents all the steps of the research in a way that would let others replicate the study. There are also additional materials available: genomic organization of dnaJ, bacterial strains, plasmid and primers used in the research.
The Funding, the Author's contribution, and Conflict of interest are correctly presented.
The list of References is made by important articles in this field, some recently published.
Based on all of these, I consider that the paper could be published after minor revisions (editing, spelling, a check for English) but not changes of the substance of the research.
Author Response
Reviewer #1 Comments and Suggestions for Authors
The authors proved that the HSP dnaJ has a role in the biofilm formation and the pathogenicity of P aeruginosa. They proved that a dnaJ mutation will not affect the survival, but will decrease the production of pyocyanine (through QS pathways), the biofilm formation (through PYO and eDNA) and the full pathogenicity of P. aeruginosa require a functional dnaJ.
The paper is well written following all steps needed to prove the hypothesis.
The "Introduction" presents the known roles of HSP, with special attention to dnaJ and P.aeruginosa. here -just a spelling error in line 2: P.aeruginosa is ability... is able
Response: We appreciate the reviewer’s positive comments on the work. The error in line 2 of page 2 has been corrected.
The results are presented in a clear order and a way that permits the reader to follow and to understand well the idea of this research. All the needed steps of the research are also proved by the figures presented and explained. The aim of the research was followed and proved clearly and the conclusions of the study are fully supported by the facts.
In the "Discussion" chapter, the authors explained clearly the results in relation to other data published in this field.
The "Material and methods" chapter presents all the steps of the research in a way that would let others replicate the study. There are also additional materials available: genomic organization of dnaJ, bacterial strains, plasmid and primers used in the research.
The Funding, the Author's contribution, and Conflict of interest are correctly presented.
The list of References is made by important articles in this field, some recently published.
Based on all of these, I consider that the paper could be published after minor revisions (editing, spelling, a check for English) but not changes of the substance of the research.
Response: We appreciate all reviewer’s positive comments above on the work and the manuscript have been checked by a native English speaking colleague.
Reviewer 2 Report
page 1 Line 31: However, It it has become apparent that the HSPs participate in gene regulation not only for protecting....
page 2 Line 2: pathogen that can cause serious infections in humans [10]. P. aeruginosa is ability able to produce a variety
page 2 Line 4-7: .... P. aeruginosa secretes many pigmented phenazine compounds that is involve in host immune response evasion [11],. and the The most well studied pigmented phenazine compound is pyocyanin (PYO); which is an electrochemically active metabolite and also a signal, involving involve in gene regulation and maintaining fitness of bacterial cells [12].
Author Response
Reviewer #2 Comments and Suggestions for Authors
page 1 Line 31: However, It it has become apparent that the HSPs participate in gene regulation not only for protecting....
page 2 Line 2: pathogen that can cause serious infections in humans [10]. P. aeruginosa is ability able to produce a variety
Response: All errors mentioned above have been corrected.
page 2 Line 4-7:.... P. aeruginosa secretes many pigmented phenazine compounds that is involve in host immune response evasion [11],. And the The most well studied pigmented phenazine compound is pyocyanin (PYO); which is an electrochemically active metabolite and also a signal, involving involve in gene regulation and maintaining fitness of bacterial cells [12].
Response: The sentences have been simplified and revised to make the description clearer.